# Correlation Analysis between Land Use/Cover Change and Air Pollutants—A Case Study in Wuyishan City

**Zhipeng Zhu** [1,2]**, Guangyu Wang** [2]  **and Jianwen Dong** [1,]*****

[1]  College of Landscape Architecture, Fujian Agriculture and Forestry University, Fuzhou 350002, China
[2]  Faculty of Forestry, The University of British Columbia, Vancouver, BC V6T 1Z4, Canada
*  Correspondence: fjdjw@fafu.edu.cn; Tel.: +86-13609525156

**Abstract:** Land use changes have significantly altered the natural environment in which humans live. In urban areas, diminishing air quality poses a large threat to human health. In order to investigate the relationship between land use/cover change (LUCC) and air pollutants of Wuyishan City between 2014–2017, an integrated approach was used by combining remote sensing techniques with a landscape ecology methods. Annual, seasonal, and weekly mean values of air pollutant ($SO_2$, $NO_2$, CO, $PM_{10}$, $O_3$, $PM_{2.5}$, black carbon) concentration and atmospheric visibility were calculated to develop a Pearson correlation between LUCC and air pollutants concentration. Results showed an increase in forested areas (1.79%) and water areas (15.89%), with a simultaneous reduction in cultivated land (6.47%), bare land (72.61%), and built-up land (16.03%) from 2014 to 2017. The transition matrix of land use types revealed that (i) forest expansion took place mainly at the expense of cultivated land (13.94%) and bare land (27.48%); and (ii) water area expansion took place mainly at the expense of cultivated land (1.29%) and forests (0.21%). In 2017, the proportion of days with AQI level I (94.52%) was higher than that in 2014 (88.77%). Additionally, the annual average visibility in 2017 (37.42 km) was higher than 2014 (27.46 km). The concentration of $SO_2$, CO, $O_3$, and black carbon was positively correlated with the cultivated land. The concentration of $SO_2$, CO, and black carbon negatively correlated with the increase of forests. $PM_{10}$, and $PM_{2.5}$ is negatively correlated with the water area. Visibility was found to be positively correlated with forested area, and negatively correlated with cultivated land. The findings from this study represent a valuable gain in understanding of policies aimed at improving, safeguarding, and monitoring air quality. These results can be used to inform land-use planning decisions in a comprehensive way and could be a valuable tool for LUCC rational management strategies.

**Keywords:** land use/cover change (LUCC); air pollution; correlation; Wuyishan City

## 1. Introduction

Since China's reform, rapid urban development has brought great benefits to its economy and society, and now China is the most industrialized country and the second largest economy in the world. However, despite these achievements, China's rapid development has also brought many social and environmental problems. With continuous development and urbanization, China's total urban area has increased along with pollution emissions from industrial activities, motor vehicles, and other human activities [1]. Atmospheric pollutants have led to various social and environmental issues, including those associated with acid rain, ozone layer depletion, human health, as well as economic, and aesthetic impacts associated with ground level air pollution [2–4]. Research has shown that air pollution has a significant impact on public health [5,6]. In recent years, issues with China's air pollution have raised

considerable concern among government officials, experts, scholars, and the general public, driving the need for further research on the subject. Pope (2000) found that long-term repeated exposure to fine particulate pollution, resulted in significant reduction in average life expectancy in highly polluted environments [7]. Chai et al. (2018) found that high $PM_{2.5}$ concentrations were associated with an increase in the daily outpatient visits for respiratory diseases [8]. Additionally, Landrigan et al. (2019) attributed the deaths of 940,000 children worldwide to air pollution in 2016 [9]. It can be seen that the air pollution problem brought about by urbanization is an urgent problem to be solved.

Many studies have shown that there is a significant relationship between LUCC and air quality. Different cities and land-use types have different effects on air quality [10,11]. Land-use can directly and indirectly affect both natural surface coverage and urban air quality. Implementation of forests and water area in urban areas can significantly improve local urban air quality [12,13]. On the contrary, land-use types, such as built-up land, can be associated with higher levels emissions of air pollution [3,14]. The idea that built-up land types have a significant impact on air quality has been proposed in the past [15,16], and has since been supported by several studies [17,18]. Zahari et al. (2016) explored the relationship between built-up land and air quality in Malaysia, and the results showed that air quality can be manipulated or reduced through proper land use planning [19]. Zou et al. (2016) conducted research at Changsha, Zhuzhou, and Xiangtan and concluded that optimizing urban LUCC can effectively improve air quality [20]. Additional studies have shown that urbanization changes the land use structure, and the increase of built-up land will lead to the strengthening of regional activities, such as living discharges caused by population intensive, transportation, etc., and these activities will directly or indirectly deteriorating air quality [21,22]; Burley et al. (2008) and Loures et al. (2016) also indicated that people pay more attention in post-industrial landscape redevelopment and public participation in urban planning process [23,24]. Land use strategies should be given heavier emphasis when formulating policies to reduce air pollution [25].

The studies mentioned above show that the LUCC has a strong correlation with the change in air quality. However, most of the previous studies only focused on the impacts of land use on air pollutants in urban built-up areas, and their study areas had complex and diverse social and economic structure [26–28]. This makes it difficult to differentiate impact factors. Wuyishan City is the only city named after by scenic spot (Wuyi Mountain) in China, which is famous all over the world. Its economy relies heavily on tourism, agriculture, as well as under-forest economic crops. Although socioeconomic factors are also considered to be a primary source of air pollution [29,30], the research geostatistical yearbook found that the socioeconomic factors for the research area were relatively stable during the study period and, therefore, these are not listed as impact factors in this study. In summary, this paper focuses on select LUCC that are closely related to changes in air quality [31,32]. In Wuyishan, secondary industry had remained stable, so the social, economic, industrial, and transportation factors had lower impacts on air pollution than other cities, and the research sites were representative. It is beneficial to study the correlation between LUCC and air pollutants. Therefore, we choose Wuyishan City as the study area to analyze the correlation between LUCC and air pollutants, in hopes to guide the planning and development of the city. This study aims to (i) study the transformation characteristics of the five types of land use (built-up land, forests, cultivated land, water area, and bare land) within the Wuyishan City from 2014 to 2017; (ii) analyze the annual, seasonal, and weekly variation characteristics of air pollutants concentration; and (iii) investigate the relationship between LUCC and air pollutants.

## 2. Materials and Methods

### 2.1. Overview of Wuyishan City

Wuyishan City is located in the northwestern part of Fujian Province, at the junction with Jiangxi Province, and is characterized as mid-subtropical climate (Figure 1). The geographical coordinates are 117°37′22″ E~118°19′44″ E and 27°27′31″ N~28°04′49″ N. The city has a total area of 2813 km², and a total population of 234,400 (2010). The economy of Wuyishan City has grown steadily, its annual

production was valued at $2.211 billion USD in 2016. Among the production value, the primary industry declined, the secondary industry continued to accelerate, and the tertiary industry developed steadily. They accounted for 16.30%, 40.52%, and 43.17%, respectively. The annual gross domestic product (GDP) was $24.180 million USD. In December 1999, it was approved by the United Nations Educational, Scientific, and Cultural Organization (UNESCO) to be included in the World Heritage List, making it one of four Chinese sites to be included in this title and one of the 23 "double heritage" of world culture and nature locations.

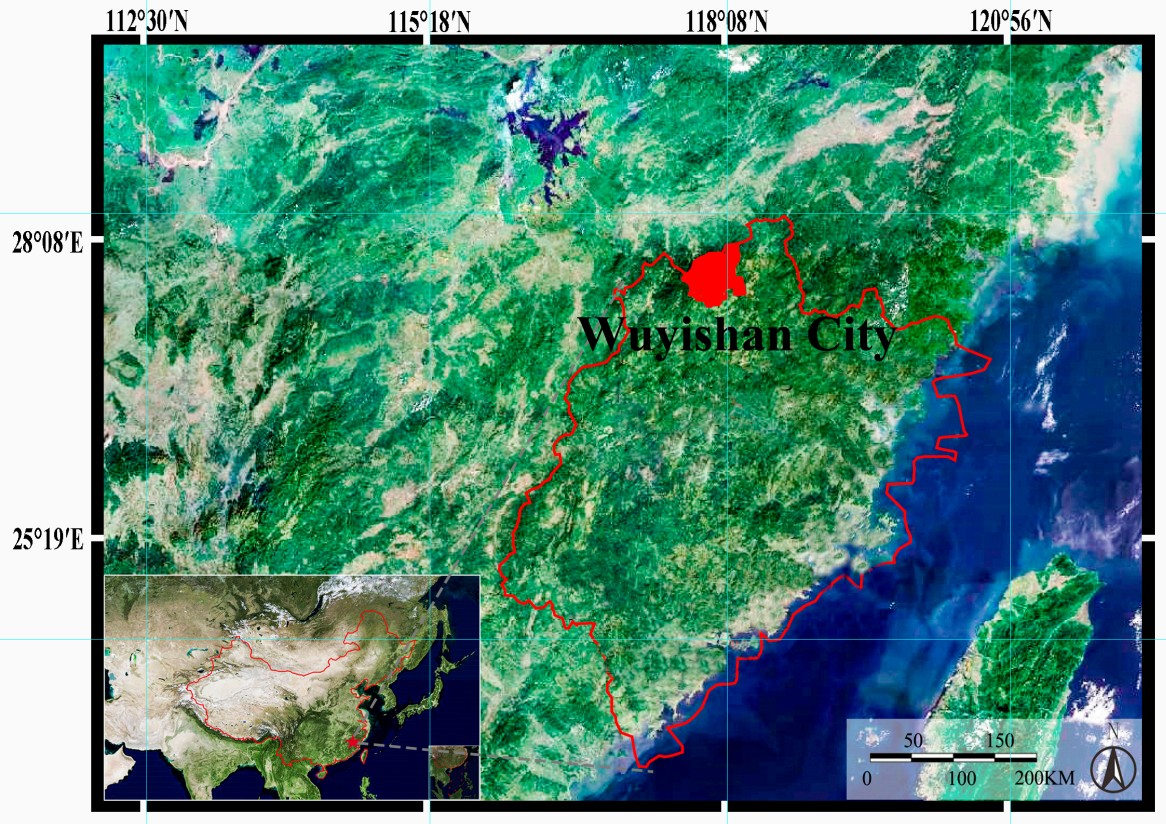

**Figure 1.** Location of study area.

*2.2. Data Sources and Research Methods*

2.2.1. Remote Sensing Image Source and Processing

Remote sensing images used for this study were selected from the National Geographic Data Cloud (http://www.gscloud.cn) (February 2014; February 2015; February 2016; and 31 March 2017). In order to minimize the extraneous variability, images were selected from the same (or as close as possible) season, period, and time. Subtle differences in image reflection characteristics caused by eliminating solar height and phenology were unavoidable [33]. Once selected, images were interpreted and analyzed using ENVI 5.3 software (Exelis Visual Information Solutions Inc, United States).

According to the Classification of Land Use Status (GB/T21010-2007), the utilization and characteristics of land resources and the characteristics of remote sensing images in Wuyishan City were combined. The land area was divided into 5 land uses types: forests, cultivated land, water area, built-up land and bare land. To perform image interpretation and analysis, a supervised classification methodology was adopted through the usage of ENVI5.3 software, for image interpretation of the remote sensing images of Wuyishan City from 2014 to 2017. The supervised classification of Wuyishan City from 2014 to 2017 was performed through the following four steps:

i.　　　the Radiometric Calibration tool of ENVI 5.3 was used to calibrate the image;

ii.　　the FLAASH Atmospheric Correction was used to perform image atmospheric correction;

iii.　　the FLAASH tool was used for parameter correction; and

iv.　　"Likelihood Classification" and "Maximum Likelihood" tools were used to supervise the classification methodology.

After interpretation, remote sensing image processing was carried out through the following 4 steps:

i.　　　Plot verification was performed using Google Earth;

ii.　　Field survey were conducted on plots with poor resolution;

iii.　　ENVI CLASSICLE software was used for secondary visual interpretation; and

iv.　　the "Confusion Matrix Using Ground Truth ROIs" tool was used for accuracy verification using a supervised inspection of approximately 2% of the total land area on the remote sensing image [34,35].

### 2.2.2. Air Pollutants Concentration Change

Air pollutant data from 2014 to 2017 was obtained from the Wuyishan monitoring station of the National Atmospheric Environment of Fujian Province (http://hbt.fujian.gov.cn/). Air pollution parameters obtained included concentrations of $SO_2$, $NO_2$, CO, $PM_{10}$, $O_3$, $PM_{2.5}$, black carbon, as well as atmospheric visibility. The annual, seasonal and weekly mean concentration of the air pollutants were calculated using Excel 2013.The Air Quality Index (AQI) was additionally calculated using Equation (1), as follows:

$$\text{AQI} = \frac{I_{high} - I_{low}}{C_{high} - C_{low}}(C - C_{low}) + I_{low} \tag{1}$$

where AQI refers to the air quality index; C refers to the instantaneous pollutant concentration; $C_{low}$ refers to the concentration limit $\leq C$; $C_{high}$ refers to the concentration limit $\geq C$; $I_{low}$ refers to the index limit corresponding to $C_{low}$; and $I_{high}$ refers to the index limit corresponding to $C_{high}$.

Following the methodology set out by the "Environmental Air Quality Standard–GB 3095-2012", air quality was divided into five levels based on the AQI: 0–50, 51–100, 101–200, 201–300, and 300–500, represented as levels I, II, III, IV, and V, respectively, on the national air quality standards GB/3095–2012. Following the methodology of previous studies, the correlation between air pollutants concentration data and LUCC was analyzed by SPSS 19.0 software to understand the relationship between pollutants and LUCC [36,37].

### 2.2.3. Correlation between LUCC and Air Pollutants

Correlations between LUCC and concentrations of $SO_2$, $NO_2$, CO, $PM_{10}$, $O_3$, $PM_{2.5}$, and black carbon, as well as atmospheric visibility were quantified based on the Pearson correlation coefficient (Equation (2)):

$$\rho_{x,y} = \frac{\sigma_{xy}}{\sigma_x \sigma_y} \tag{2}$$

where $\rho_{x,y}$ represent the covariance between the two variables, $\sigma_x$ represent the standard deviation of land use type, and $\sigma_y$ represent the standard deviation of pollutants. The Pearson correlation coefficient was used to measure the degree of correlation between the two variables. A correlation value of 1 indicated that the variables were significant positively correlated, a value of −1 indicated a significant negative correlation between the variables, and a value of 0 indicated no significant relationship between the variables [38].

## 3. Results

### 3.1. LUCC in Wuyishan City

The classification results for the 2014 and 2017 periods are shown in Figure 2. The LUCC transfer matrix is presented in Table 1. From 2014–2017, forestry and water area land area increased, and cultivated land, bare land, and built-up land area decreased. Cultivated land area decreased considerably over this time period from 242.512 km² to 226.833 km² at a rate of 6.47% from 2014 to 2017. The characteristics of two-way changes in transfer are outstanding. The forest area change rate was the lowest at 1.79% from 2014 to 2017; however, due to its large base, this accounted for a total area change of 44.438 km². Over the study period, the water area was increased by 2.861 km² in total with a change rate of 15.89%. Built-up land area changed with a total decrease in area of 5.822 km², and a rate of change of 16.03%. The newly built-up land area mainly transferred from cultivated land and bare land and was mostly concentrated on existing towns. Bare land had the highest rate of change at 72.61% with a total area reduction 25.77 km². The decrease in Bare land area could mainly be attributed to the conversion into agricultural and forestry.

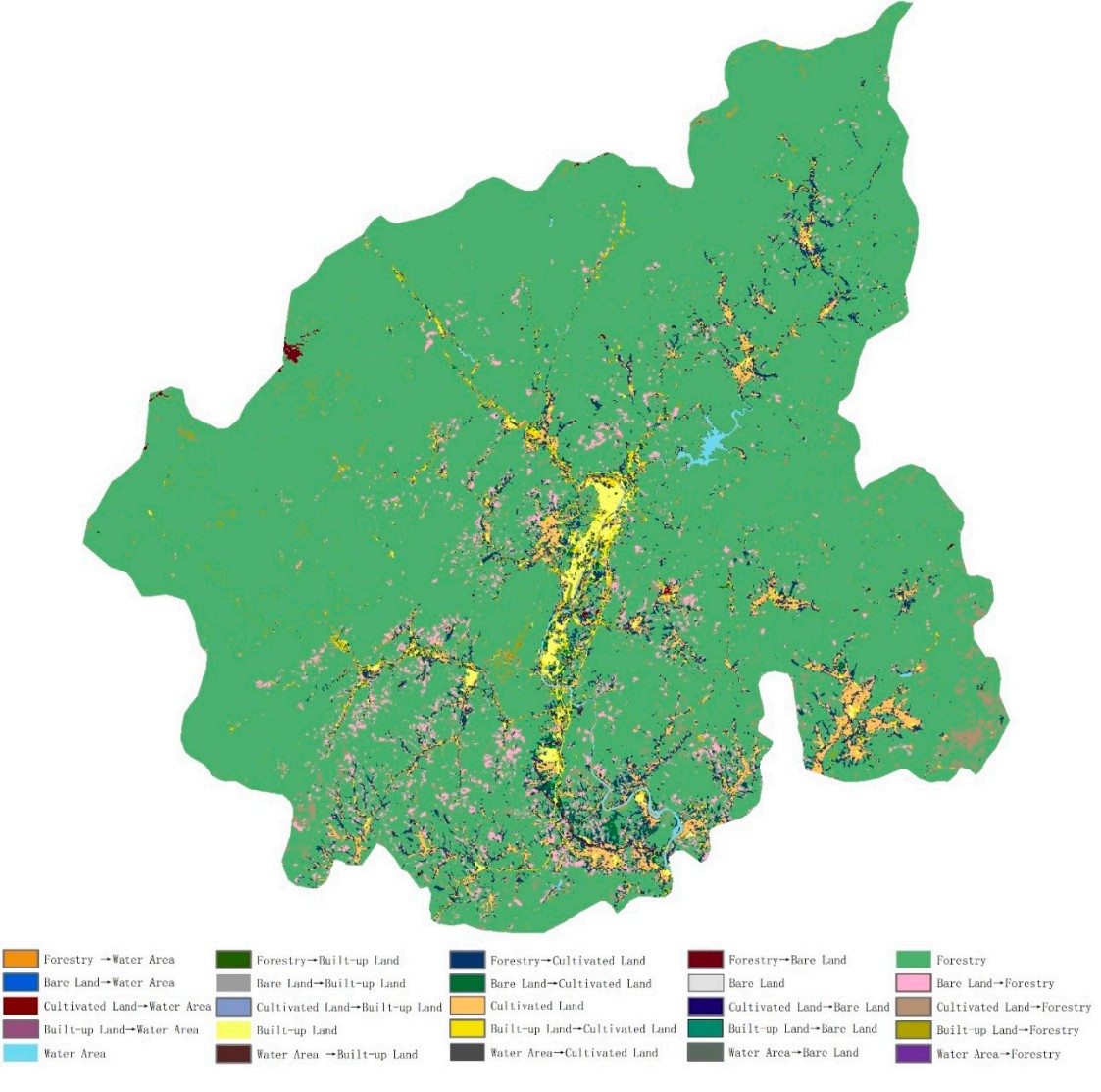

**Figure 2.** Land use change of 2014–2017.

**Table 1.** Matrix of land use change during 2014–2017 ($km^2$).

| Land Use Type | Cultivated Land ($km^2$) | Bare Land ($km^2$) | Water Area ($km^2$) | Built-Up Land ($km^2$) | Forestry ($km^2$) | 2017 Land Use ($km^2$) |
|---|---|---|---|---|---|---|
| Cultivated land ($km^2$) | 172.901 | 4.514 | 3.020 | 12.581 | 33.795 | 226.810 |
| Bare land ($km^2$) | 2.957 | 2.736 | 0.130 | 0.374 | 3.524 | 9.721 |
| Water area ($km^2$) | 3.124 | 0.290 | 11.395 | 0.781 | 5.271 | 20.861 |
| Built-up Land ($km^2$) | 7.108 | 0.227 | 0.185 | 21.33 | 1.650 | 30.502 |
| Forest ($km^2$) | 56.410 | 27.718 | 3.272 | 1.253 | 2434.567 | 2523.220 |
| 2014 land use ($km^2$) | 242.512 | 35.492 | 18.001 | 36.324 | 2478.908 | – |
| Class Changes ($km^2$) | 69.611 | 32.756 | 6.606 | 14.991 | 44.341 | – |
| Image Difference ($km^2$) | −15.679 | −25.770 | 2.861 | −5.822 | 44.438 | – |

*3.2. Concentration of Air Pollutants*

In 2014, a total of 324 days had an AQI of level I, and 50 days had an AQI level of II (Table 2). In 2015, the number of days with an AQI level I was increased to 339 days. In 2016, the number of days with an AQI level I was highest from 2014–2017, reached 356 days. In 2017, the number of days with an AQI level I was 345 days, whereas the number of AQI level II days was only 16 (Table 2). Table 2 also indicates that in 2017, a pollution event took place over a total of four days where the AQI reached level III. The specific data and variation characteristics are shown in Table 2. Overall, the data showed that Wuyishan City has great air quality, and there were no serious pollution problems. The results also showed that the air quality in 2016 and 2017 was better than 2014 and 2015. As shown in Figure 3, the concentration of various pollutants in 2014 and 2015 was higher than that in 2016 and 2017. The concentration of $SO_2$, $PM_{10}$ and $NO_2$ differs greatly, and the concentration of CO and $O_3$ are similar. In addition, the visibility level in 2014 was lower than 2017, further indicating that the air quality of Wuyishan City had increased from 2014 to 2017 (Figure 3).

**Table 2.** Percentage of urban air quality standards for 2014–2017 in Wuyishan City.

| Standards of Air Quality | I | II | III | IV | V |
|---|---|---|---|---|---|
| 2014 | 88.77% | 11.23% | 0.00% | 0.00% | 0.00% |
| 2015 | 92.88% | 7.12% | 0.00% | 0.00% | 0.00% |
| 2016 | 97.27% | 2.73% | 0.00% | 0.00% | 0.00% |
| 2017 | 94.52% | 4.38% | 1.10% | 0.00% | 0.00% |

Figure 3 and Table 3 showed that average concentrations of most air pollutants were higher in 2014. Air pollutants concentration value were similar in 2016 and 2017. The air pollutants concentration showed a gradual decline from 2014 to 2017. Therefore, the visibility was higher in 2017 than 2014. We also found that the air quality in spring and winter was lower than summer and autumn. In 2017, the concentration of air pollutants in summer, autumn and winter were lower than those in 2014, and visibility levels were higher than those in 2014 (Table 3). The results show that concentrations of all air pollutants were higher in 2014 than in 2017. From 2014 to 2017, $NO_2$, CO, $O_3$, $PM_{10}$, $PM_{2.5}$, and black carbon decreased in autumn and visibility increased in summer (Table 3). This indicates that, the overall air quality was higher in 2017 than 2014.

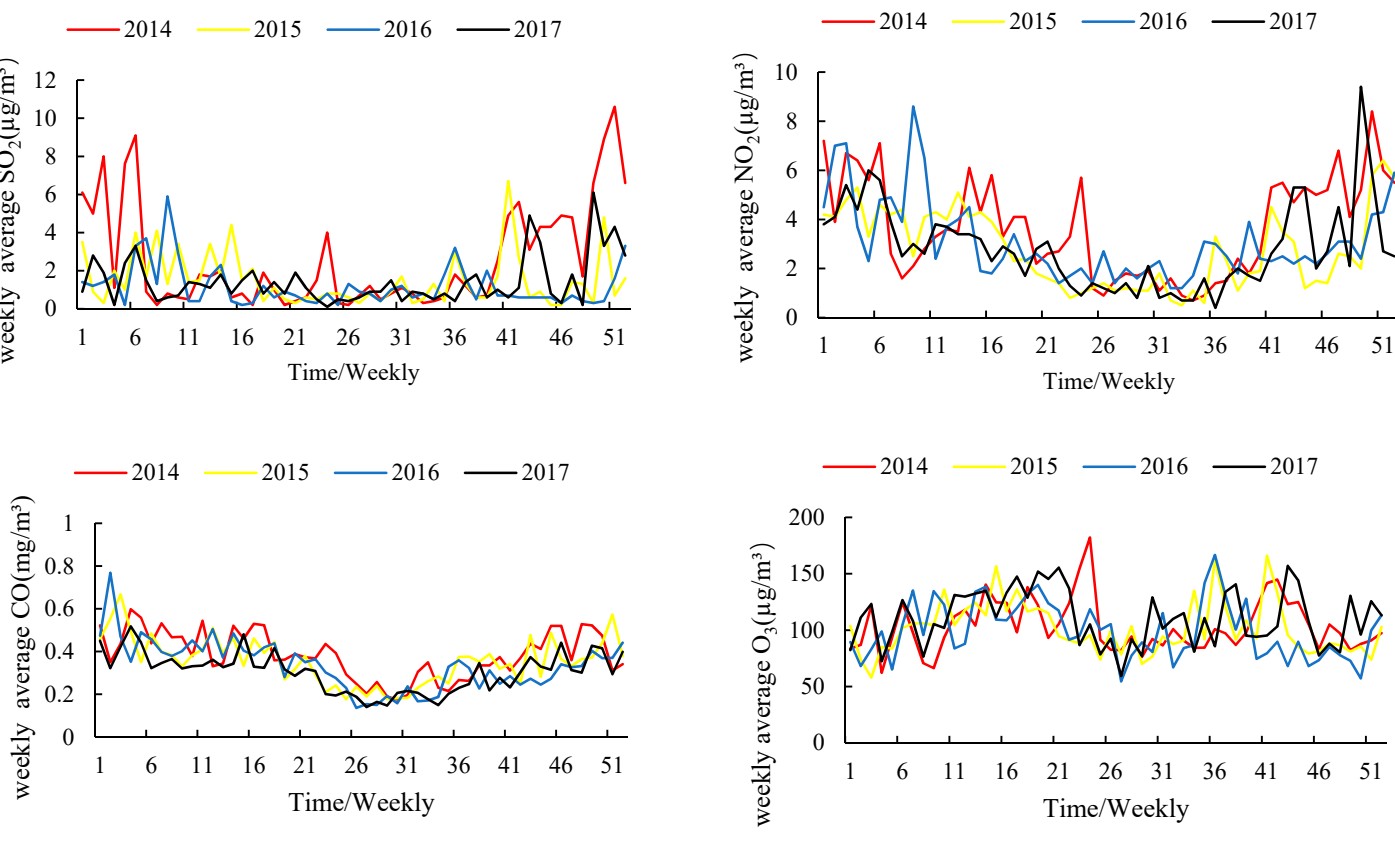

**Figure 3.** *Cont.*

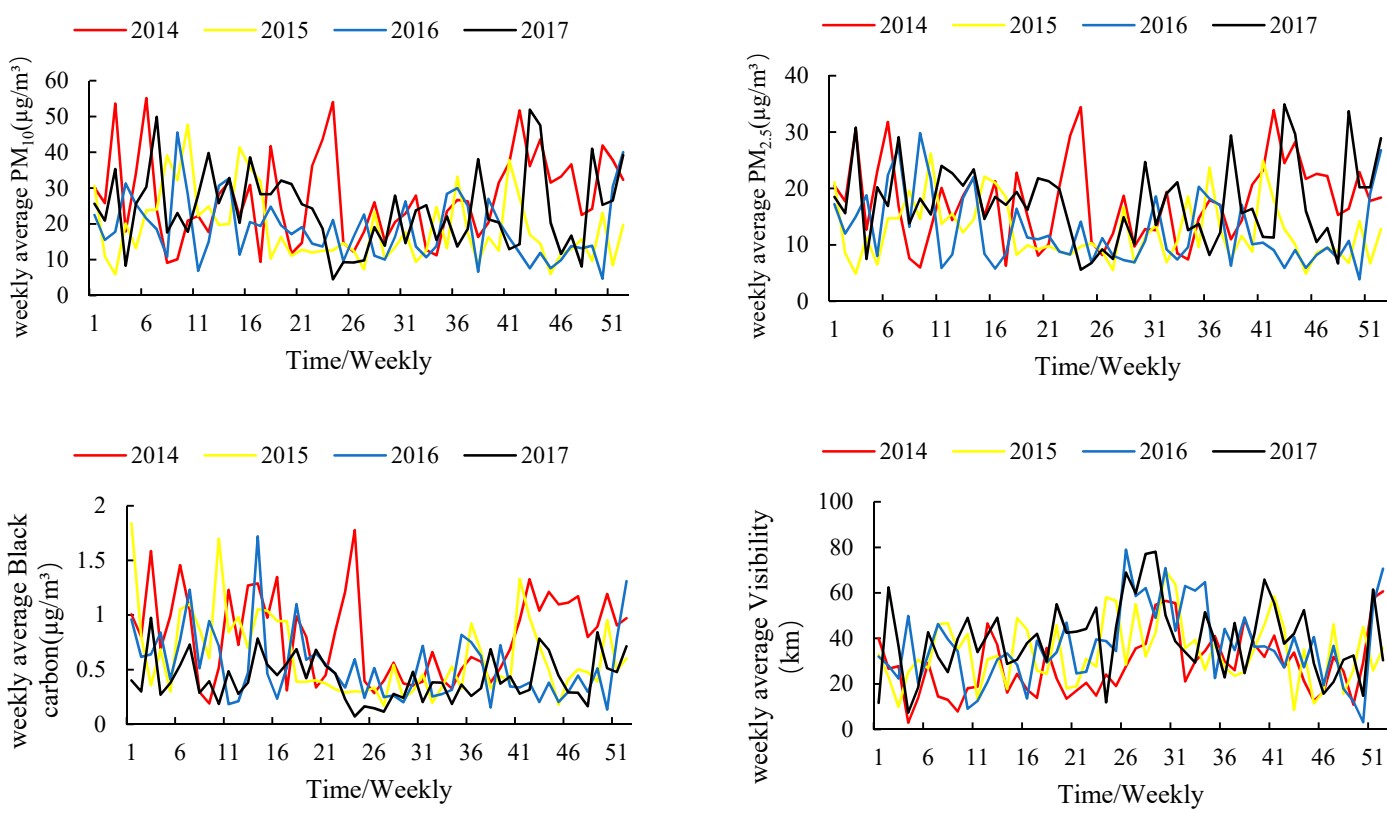

**Figure 3.** Weekly averages for $SO_2$, $NO_2$, CO, $PM_{10}$, $O_3$, $PM_{2.5}$, and black carbon concentration, visibility from 2014–2017.

**Table 3.** Seasonal variation characteristics of air pollutant concentration from 2014 to 2017.

| Air Pollutants | Spring | | | | Summer | | | | Autumn | | | | Winter | | | | Annual | | | |
|---|---|---|---|---|---|---|---|---|---|---|---|---|---|---|---|---|---|---|---|---|
| | 2014 | 2015 | 2016 | 2017 | 2014 | 2015 | 2016 | 2017 | 2014 | 2015 | 2016 | 2017 | 2014 | 2015 | 2016 | 2017 | 2014 | 2015 | 2016 | 2017 |
| $SO_2$ ($\mu g/m^3$) | 0.85 | 1.49 | 0.75 | 1.71 | 0.95 | 0.91 | 1.04 | 0.72 | 3.11 | 1.42 | 0.74 | 0.96 | 5.89 | 2.31 | 2.18 | 1.68 | 2.72 | 1.52 | 1.19 | 1.20 |
| $NO_2$ ($\mu g/m^3$) | 3.91 | 3.02 | 2.64 | 3.63 | 1.79 | 1.24 | 1.97 | 1.22 | 3.97 | 2.26 | 2.58 | 2.62 | 5.52 | 4.48 | 5.21 | 4.78 | 3.73 | 2.71 | 3.05 | 3.21 |
| CO ($\mu g/m^3$) | 0.43 | 0.38 | 0.39 | 0.38 | 0.27 | 0.23 | 0.21 | 0.20 | 0.39 | 0.36 | 0.29 | 0.29 | 0.46 | 0.46 | 0.45 | 0.44 | 0.39 | 0.36 | 0.33 | 0.34 |
| $O_3$ ($\mu g/m^3$) | 118.93 | 114.95 | 113.98 | 128.54 | 100.22 | 96.88 | 99.04 | 98.66 | 109.85 | 100.85 | 87.59 | 94.81 | 91.68 | 93.18 | 97.55 | 91.87 | 103.76 | 101.63 | 100.60 | 99.39 |
| PM10 ($\mu g/m^3$) | 24.74 | 20.82 | 18.86 | 27.93 | 23.24 | 15.76 | 17.70 | 16.69 | 31.79 | 15.98 | 14.25 | 15.49 | 32.16 | 23.25 | 24.02 | 19.71 | 27.38 | 18.87 | 18.12 | 18.40 |
| PM2.5 ($\mu g/m^3$) | 15.67 | 13.21 | 11.12 | 18.16 | 15.24 | 11.70 | 11.42 | 13.77 | 20.96 | 11.16 | 9.76 | 10.34 | 19.68 | 13.58 | 18.08 | 15.41 | 17.64 | 12.39 | 12.33 | 12.60 |
| Black carbon ($\mu g/m^3$) | 0.81 | 0.67 | 0.58 | 0.55 | 0.59 | 0.38 | 0.43 | 0.26 | 0.88 | 0.56 | 0.38 | 0.40 | 0.99 | 0.90 | 0.76 | 0.68 | 0.82 | 0.62 | 0.52 | 0.52 |
| Visibility (km) | 20.13 | 29.09 | 29.08 | 31.19 | 34.77 | 44.05 | 53.21 | 50.62 | 29.52 | 30.26 | 31.80 | 34.25 | 27.22 | 32.44 | 34.10 | 35.22 | 28.69 | 33.99 | 37.38 | 37.24 |

### 3.3. Relationship between LUCC and Air Pollutants

The Pearson correlations between LUCC and air pollutant concentration from 2014–2017 was displayed in Table 4. The results show that concentrations of $SO_2$, CO, $O_3$, and black carbon were all positively correlated with the cultivated land. Similarly, concentrations of $SO_2$, CO, and black carbon were negatively correlated with forestry. $PM_{10}$ and $PM_{2.5}$ were negatively correlated with the water area. There were no correlations between air pollutant concentrations and built-up land or bare land. Atmospheric visibility was found to be positively correlated with forestry area, and negatively correlated with cultivated land. These indicated that visibility similarly improved with the increase in forestry area (Table 4).

**Table 4.** Correlation between air pollutant concentrations and LUCC.

| Air Pollutants Concentration | LUCC | | | | |
|---|---|---|---|---|---|
| | **Water Area** | **Built-Up Land** | **Cultivated Land** | **Bare Land** | **Forestry** |
| $SO_2(\mu g/m^3)$ | −0.944 | −0.241 | 0.965 * | 0.767 | −0.964 * |
| $NO_2(\mu g/m^3)$ | −0.872 | −0.621 | 0.558 | 0.794 | −0.575 |
| CO (mg/m$^3$) | −0.929 | −0.308 | 0.968 * | 0.825 | −0.987 * |
| $O_3(\mu g/m^3)$ | −0.768 | 0.112 | 0.972 * | 0.509 | −0.945 |
| $PM_{10}(\mu g/m^3)$ | −0.966 * | −0.328 | 0.915 | 0.799 | −0.917 |
| $PM_{2.5}(\mu g/m^3)$ | −0.971 * | −0.376 | 0.880 | 0.813 | −0.884 |
| Black carbon ($\mu g/m^3$) | −0.912 | −0.179 | 0.991 ** | 0.740 | −0.991 ** |
| Visibility(km) | 0.897 | 0.156 | −0.995 ** | −0.726 | 0.996 ** |

Note: ** means the correlation is significant at the 0.01 level (two-tailed); * means significant at the 0.05 level (two-tailed).

## 4. Discussion

### 4.1. The LUCC between 2014 and 2017

The interpretation and analysis of LUCC in Wuyishan City showed that built-up land was relatively stable between 2014 and 2017. This could be attributed to afforestation and restrictions on urban development. Over this period, significant land conversion into forests occurred, resulting in an increase of forest coverage [39]. This effect is likely in response to current policy which focused on restoring cultivated land to forests in China [40–42]. Fan et al. (2019) further confirmed that the forested areas in the Nanping City has increased, while the area of cultivated land has decreased from 2010 to 2015 [43]. In order to meet the need of the tourism and tea industry in Wuyishan City, most of bare land had turn into forests (78.10%) and cultivated land (12.72%). The study of Wuyishan tourism by Chen et al. (2017) are consistent with our findings, residents in Wuyishan appeared supportive of the conservation of natural resources and local culture [44]. Wu et al. (2016) revealed that LUCC and urban expansion were highly correlated with economic development, population growth, technical progress, policy elements, and other similar factors [45]. From the perspective of LUCC, forest and cultivated land were the largest land use type in Wuyishan, which relates to the economic structure of Wuyishan. This finding was consistent with Yang et al. (2011), whereby tourism has continued to develop, and the tea industry has remained stable in Wuyishan [46]. However, in Northeast China, in order to meet the needs of economic development, population growth, and industrialization, many cities have relied on continuous expansion which is leads to the increased built-up of land area [47–49]. The increasing demand for built-up areas caused the disappearance of a large amount of cultivated land and forests surrounding the cities [50]. During 2014 and 2017, the change in water area was relatively stable, mainly due to increased investment in water conservation, environmental protection of water ecology, and energy conservation in China [46].

### 4.2. Change Characteristics of Air Pollutants in 2014–2017

The results indicate that the overall air quality showed great condition in Wuyishan, more than 88.77% of the days in one year reached the level I. This is mainly related to the key development of forestry-related industries in Wuyishan City. This result was consistent with the study by Francisco et al. (2009) on the effects of urban forest on spatial heterogeneity and air pollution in Santiago de Chile, and found that air quality improvement was mainly related to vegetation [51]. Our study found that there were similar changes among air pollutants. Allen's (1990) research corroborates our results, indicated that there was a potential synergy between air pollutants [52]. The overall air quality and atmospheric visibility were better in 2017 than 2014 mainly due to the increased of forest area. This finding was verified by Irga et al. (2015), which found that air samples taken from sites with less greenspaces frequently had higher concentrations of all fractions of aerosolized particulates than sites with high proximal greenspaces, which had lower particulates [53]. In the spring of 2017, the concentrations of $SO_2$, $PM_{10}$ and $PM_{2.5}$ were higher than that in the spring of 2014. The slight pollution phenomenon appeared in Wuyishan City, which may be related to the release of fireworks and firecrackers during the Spring Festival. Studies have shown that the fireworks and firecrackers could release a large amount of $SO_2$ and fine dust, which leads to an increased in the concentrations of $SO_2$, $PM_{10}$, $PM_{2.5}$ [51,54]. This pollution phenomenon seems to be consistent with this idea since the increase in pollutants was only observed in spring, and pollutant concentrations were lower in all other seasons in 2017.

### 4.3. Relationships between LUCC and Air Pollutants in the Reference Period

Our research found that $SO_2$, CO, $O_3$, and black carbon concentration were positively correlated with cultivated land. In contrary, there was a significant negative correlation between $SO_2$, CO, black carbon and the increase of forested areas (Table 4). This indicates that an increase in cultivated land areas would reduce air quality, and the increase in forested areas would improve air quality by affecting the air pollutants. This finding was consistent with Huang et al. (2018), which examined the effects of urban land expansion on air pollution in China, and found that urban land expansion influenced air pollution, with a significant relationship to the concentration of air pollutants [55]. Zhang et al. (2015) found that visibility was significantly affected by the variation of PMs concentrations [56], and many studies have also shown that forests can effectively reduce the concentration of PMs [15,29]. Therefore, the increase in the forest can effectively improve visibility, which was consistent with the results of this research. It also indicated that the atmospheric environment quality of Wuyishan City has been further improved. The variation of air pollutants in Wuyishan City is related to various factors, including city area expansion, industrial activities, population, meteorological factors, policies, urban forest, and so on. As forest coverage increases, it can absorb more pollutants and improve air quality [57]. Rodriguez et al. (2016) investigated the relationship between local air pollution and urban structure with an emphasis on urban fragmentation, and their findings support the hypothesis that urban structure has a significant impact on air pollutant concentration [58]. These existing observations were also supported by our findings [59–61]. Due to the conversion of cultivated land into forests in Wuyishan City, average carbon sequestration and oxygen release value has significantly increased, which reduces the concentration of air pollutants [62]. This is consistent with previous research indicating that a decrease in built-up land, and an increase in forests and green area, led to a lower concentration of $NO_2$ and $PM_{2.5}$ [58,63,64]. Our research results showed that the correlation between bare land and air pollutants is poor. The reason may be due to bare land mainly turned into cultivated land and forestr, and our research found that cultivated land is positively correlated with many air pollutants [55], while forestry land is negatively correlated with many air pollutants [59], which leads to the insignificant correlation between bare land and air pollutants. Built-up land and air pollutants also showed no significant. Our research found that the change in built-up land during the study period was less obvious and remained at a relatively stable condition, which may be the main reason for the inconspicuous results of our correlation test.

*4.4. Application: Policy Recommendation*

China's rapid economic development has resulted in many environmental problems. In order to protect and repair these environmental problems, the Chinese government has integrated the construction of ecological civilization into the overall layout of the socialist cause with Chinese characteristics. For example, in 2014 the Ministry of Land and Resources of China issued the "Regulations on the Use of Land for Saving and Intensive Use" policy, which restricted the supply of new built-up land in 16 megacities undergoing urban construction and development. This shows that China is beginning to prioritize the protection of ecology and environment in urban development. As stated in 4.3, the restrictions on the supply of built-up land can effectively avoid further deterioration of the environment. The findings of this study suggest that more attention should be paid to the construction of urban green spaces and urban forests. Future development processes should consider the importance of the protection of forest and water area. Additionally, the relationship between built-up land, cultivated land, bare land, forest and water area should be rationally planned. The results of this study indicate that regulators and city planners should limit disorderly expansion, consider ecological protection, and adhere to the urban planning model of sustainable development. In order to establish a healthy and sustainable development in Wuyishan City. We propose to combine the strong environmental regulation policy, improve market competition mechanism, and make long-term strategic planning of green development. This research found that LUCC has an impact on air quality, and the increase in forests can effectively improve the air environment of the city.

## 5. Conclusions

This paper analyzed the relationship between urban LUCC and air quality in the city of Wuyishan. The results showed that from 2014 to 2017, built-up land keep stability, and cultivated land and bare land area decreased, while forest and water area increased. The air quality measured in 2017 was better than in 2014. The relationship between LUCC and air pollutant concentrations revealed that the cultivated lands were significantly positively correlated with $SO_2$, CO, $O_3$, and black carbon concentration, while forested areas were significantly negatively correlated with $SO_2$, CO, and black carbon. Furthermore, the change in build-up land during the study period was less obvious and remained at a relatively stable condition, which may be the main reason for the inconspicuous results of our correlation test.

The results of this study indicate that changes in land uses may have direct or indirect impacts on the air quality. This research explains the correlation between LUCC and air pollutants concentration in Wuyishan City. It provides a references for regional development planning and can better improve urban planning and construction by comparing the relationship between LUCC and air pollutants for different urbanization levels in the future. In order to improve the urban air environment and enable people to live in a healthier environment, the actions of afforestation should continue to be maintained. There were also some shortcomings of this study. For example, in this study, only 5 types of land use types were classified. For future research, it is necessary to refine the classification of land types in order to guide the city's construction direction more comprehensively and accurately. The investigation of the slight pollution phenomenon in Wuyishan City in 2017 can only help make a preliminary judgment; the time span of the study can be extended appropriately if needed. Additionally, it will continue to be strengthened in future researches.

**Author Contributions:** Conceptualization: Z.Z. and J.D.; methodology: J.D.; software: Z.Z.; resources: G.W.; data curation: Z.Z.; writing—original draft preparation: Z.Z.; writing—review and editing: G.W.

**Funding:** This research was funded by Public Interest Program of Chinese Ministry of State Forestland (NO.201404301; 201404315), and Fujian Agriculture and Forestry University that supports doctoral students to go abroad to be jointly trained. This research was supported by the Fujian Agriculture and Forestry University "PhD student Study Abroad Research Program".

**Acknowledgments:** We thank the Geospatial Data Cloud and Ministry of Environmental Protection of China for the dataset. Thank Brewer Julia, Xiong Yao, Dehui Geng, and Ernest Wu's edit for this research.

**Conflicts of Interest:** The authors declare no conflict of interest.

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
