# Peer review of "Correlation Analysis between Land Use/Cover Change and Air Pollutants—A Case Study in Wuyishan City"

_energies, doi:10.3390/en12132545_

Round 1

Reviewer 1 Report

Dear authors,

Thank you very much for your manuscript.

There are several comments:

1.       English should be edited. There are many mistakes in English writing.

2.       Introduction. The literature review is not logical enough. For example, line 48 to line 50, “In recent year, issues with China’s air pollution have raised considerable concern among government offcials,……”, the literature review of China’s air pollution issue should follow this sentence.

3.       Materials and methods. The overview of Wuyishan City is too simple. The socio-economic factors should also be introduced.

4.       In the result part, 3.3 Relationship between LUCC and air pollutants. The result of Table 4 is not convincible, because even without this table, this result can be estimated. The number of correlation relationship should be specific. Just -1 and 1 is too general.

5.       Discussion. The authors cited many related studies in the discussion. However, the discussion of the results in this study is quite small. The discussion part in this paper is almost like literature review.

6.       Policy recommendation is not deep enough.

7.       What is the theory contribution in this study?

Author Response

Comment 1: English should be edited. There are many mistakes in English writing.

Response 1: Thank you for pointing out the mistake. We have edited the English at whole paper. Please see the revised manuscript. Thank you.

Comment 2: Introduction. The literature review is not logical enough. For example, line 48 to line 50, “In recent year, issues with China’s air pollution have raised considerable concern among government offcials,……”, the literature review of China’s air pollution issue should follow this sentence.

Response 2: Thank you for this valuable suggestion. We reorganized the literature review in the paper, please see lines 43-45, 52-53.

Comment 3: Materials and methods. The overview of Wuyishan City is too simple. The socio-economic factors should also be introduced.

Response 3: Thank you for this helpful suggestion. We have added the socio-economic factors about Wuyishan city in the paper. Please see line 91-94.

Comment 4: In the result part, 3.3 Relationship between LUCC and air pollutants. The result of Table 4 is not convincible, because even without this table, this result can be estimated. The number of correlation relationship should be specific. Just -1 and 1 is too general.

Response 4: Thank you for this valuable suggestion. In order to make the result more convincible, we analysed the correlation between LUCC and air pollutants from 2014 to 2017.

Please see lines 191-198, and Table 4.

Comment 5: Discussion. The authors cited many related studies in the discussion. However, the discussion of the results in this study is quite small. The discussion part in this paper is almost like literature review.

Response 5: Thank you for pointing out the mistake. In this part, we have strengthened the discussion of the results, and reorganized the Discussion. Please see lines 202-284. Thank you.

Comment 6: Policy recommendation is not deep enough.

Response 6: Thank you for this valuable suggestion. We have added the Policy recommendation in the paper. Please see lines 279-282. Thank you so much.

Comment 7: What is the theory contribution in this study?

Response 7: Thank you for this helpful suggestion. We have further explained the contribution of this study in Part 4.4. Please see lines 297-299. Thank you so much.

Thank you again for your helping and all the valuable suggestions. We really appreciate your feedback.

Reviewer 2 Report

Even if the paper addresses an interesting topic, there are several issues that need to be strengthened/reviewed, namely:

Literature review misses several seminal works and important advances crossing lad use change and poluttion. Read for example (Burley et. al., Loures et al. mainly regarding this subject in association with mining and other industrial sites).

Material and methods presents several flaws, considering not only that it is hard to understand the used methodology, but also because the research steps are not adequately described. I would recommend the introduction of a phased methodological diagram with an adequate description of each step.

The results are also not adequately presented. Figures 2 and 3 are hard to read and present little information regarding the objectives of the research - in fact I do not think anyone can find a significative difference between those images. Further information is needed and the relation between land-use change and the presence of air pollutants need to be supported on further data.

Conclusions need to be more scientific. 

As they are, they highlight existing limitations and emphasise the empirical nature of the research.

Author Response

Comment 1: Literature review misses several seminal works and important advances crossing lad use change and poluttion. Read for example (Burley et. al., Loures et al. mainly regarding this subject in association with mining and other industrial sites).

Response 1: Thank you for this helpful suggestion. We have cited the reference in this research. Please see lines 66-67. Thank you so much.

Comment 2: Material and methods presents several flaws, considering not only that it is hard to understand the used methodology, but also because the research steps are not adequately described. I would recommend the introduction of a phased methodological diagram with an adequate description of each step.

Response 2: Thank you for this helpful suggestion. We have modified it in the paper, please see lines 116-128.

Comment 3: The results are also not adequately presented. Figures 2 and 3 are hard to read and present little information regarding the objectives of the research - in fact I do not think anyone can find a significative difference between those images. Further information is needed and the relation between land-use change and the presence of air pollutants need to be supported on further data.

Response 3: Thank you for this valuable suggestion. We have combined Figures 2 and 3 into a new figure, so that readers can better see the changes in land use between 2014 and 2017 in Wuyishan. Please see Figure 2. Also, we have added further information to analyze the relation between land-use change and air pollutants. Please see Table 2 and 4. Thank you so much.

Comment 4: Conclusions need to be more scientific.

As they are, they highlight existing limitations and emphasise the empirical nature of the research.

Response 4: Thank you for this valuable suggestion. We have modified the conclusions in the paper. Please see lines 304-306.

Thank you again for your helping. We really appreciate your feedback.

Reviewer 3 Report

The organisation and presentation of the paper are correct, but the methodology is inappropriate, as are the data employed in the analysis of atmospheric pollution. I understand the main problems to involve:

1) The correlation between the different pollutants and the land cover and uses is not sufficiently explanatory, because the two variables used involve only two cases each, and under these conditions any pair of variables that have undergone a change between the two dates used would present the maximum correlation, with 0 errors.

 2) The data on air quality are from one single station and are extrapolated to an area of approximately 3000 km2, with no correction for topography or the habitual climatic conditions, such as the frequency of phases of atmospheric stability or direction of the prevailing winds; no consideration is given, either, to the theoretical specific emission of pollutants associated with each of the land uses defined.

3) There is no analysis of the variability in emissions from 2014 and 2017 in relation to the mean of the series, in order to rule out the cause of the emissions being fortuitous.

I therefore believe that the manuscript ought not to be published; perhaps you could consult another reviewer.

Author Response

Comment 1: The correlation between the different pollutants and the land cover and uses is not sufficiently explanatory, because the two variables used involve only two cases each, and under these conditions any pair of variables that have undergone a change between the two dates used would present the maximum correlation, with 0 errors.

Response 1: Thank you for this helpful suggestion. We have added 2 periods data (2015, 2016) to explain the correlation between the air pollutants and the land cover and land uses. Please see the revised manuscript. Thank you for your useful advice.

Comment 2: The data on air quality are from one single station and are extrapolated to an area of approximately 3000 km2, with no correction for topography or the habitual climatic conditions, such as the frequency of phases of atmospheric stability or direction of the prevailing winds; no consideration is given, either, to the theoretical specific emission of pollutants associated with each of the land uses defined.

Response 2: Thank you for this helpful suggestion. We understand that more weather monitoring station can make the research of the paper more scientific, but Wuyishan City has only one authoritative weather monitoring station, so this article can only provide this data. We feel really sorry about that. However, this weather station is an authoritative monitoring point, and all data is also recognized by government units and research institutes. To compensate for this deficiency, we have added additional air pollutant data (2016, 2016) for analysis.

In future research, we will strengthen the collection of data, and conduct targeted research on the characteristics of different air pollutants associated with each of the land uses defined. Please see the revised manuscript. Thank you for your wise advice.

Comment 3: There is no analysis of the variability in emissions from 2014 and 2017 in relation to the mean of the series, in order to rule out the cause of the emissions being fortuitous.

Response 3: Thank you for this valuable suggestion. We draw from the statistical yearbook data of Wuyishan City that during the period of 2014-2015, the development of the secondary industry in Wuyishan City tends to be stable, so we did not list industrial emissions as our research points. We have added this description in lines 76-82. Thank you for your wise advice.

Thank you again for your helping. We really appreciate your feedback.

Reviewer 4 Report

The article develops an analysis of the effect of land uses on air quality in a specific city/region in China. The article is sound, well written and deals with an important topic that has concern Chinese and international community for years, that is the effects of rapid industrialization and urbanization in China.

The paper can however be improved in some aspects such as:

L26 The negative correlation between LUCC and visibility is not sufficiently explained.

L64 The word “construction” is used as constructed or build, consider using “urban” or “built-up land” instead.

L66 Is it “deteriorating air pollution” or “deteriorating air quality”?

L73 “scenic area” please elaborate.

L75 “forest economic crops” please explain what is it.

L76-78 This “the research geostatistical yearbook found that the socio-economic factors for the research area were relatively low during the study period, and therefore these are not listed as impact factors in this study” What does this means? Are Socioeconomic factors irrelevant or are you saying that the variation writhing the years under study are irrelevant? In L80 authors even mention that industry has weaken. It is hard to believe that socioeconomic factors have a “relatively low” contribution. Latter on authors mention that there has been a swift between agricultural activities (tea production) to services (tourism), isn’t it inconsistent?

L81-82 if in Wuyishan “industry had been weakened, so the social, economic, industrial, and transportation factors had lower impacts on air pollution than other cities” how can authors claim that the research site is representative?

Table 1 It is not clear whether data in cells belong to 2014 or to 2017. Also the use of the dot “.” is not clear, as the number of digits on the right side of the dot is sometimes 2, 3 or 4 digits.

Author Response

Comment 1: L26 The negative correlation between LUCC and visibility is not sufficiently explained.

Response 1: Thank you for this helpful suggestion. We have explained it in the paper. Please see lines 28-29. Thank you so much.

Comment 2: L64 The word “construction” is used as constructed or build, consider using “urban” or “built-up land” instead.

Response 2: Thank you for this helpful suggestion. We have used “built-up land” instead in the paper. Please see revised manuscript. Thank you so much.

Comment 3: L66 Is it “deteriorating air pollution” or “deteriorating air quality”?

Response 3: Thank you for this valuable suggestion. It’s “deteriorating air quality”, we have corrected the errors in the paper. Please see line 65. Thank you so much.

Comment 4: L73 “scenic area” please elaborate.

Response 4: We apologize for this mistake. We have elaborated it in the paper. Please see lines 74-75. Thank you so much.

Comment 5: L75 “forest economic crops” please explain what is it.

Response 5: Thank you for this suggestion. “under-forest economic crops” refers to the planting of some valuable plants on the forest floor, which can increase the economic income of residents. This form is also widely used and promoted in China. Thank you so much.

Comment 6: L76-78 This “the research geostatistical yearbook found that the socio-economic factors for the research area were relatively low during the study period, and therefore these are not listed as impact factors in this study” What does this means? Are Socioeconomic factors irrelevant or are you saying that the variation writhing the years under study are irrelevant? In L80 authors even mention that industry has weaken. It is hard to believe that socioeconomic factors have a “relatively low” contribution. Latter on authors mention that there has been a swift between agricultural activities (tea production) to services (tourism), isn’t it inconsistent?

Response 6: We apologize for this mistake. We have modified it in the paper. Please see lines 78-81. Thank you so much.

Comment 7: L81-82 if in Wuyishan “industry had been weakened, so the social, economic, industrial, and transportation factors had lower impacts on air pollution than other cities” how can authors claim that the research site is representative?

Response 7: Thank you for this helpful suggestion. In the research, we focus on the relationship between land use change and air pollution. When the social, economic, industrial, and transportation factors had lower impacts on air pollution, it’s more convincing to explain the relationship between land use change and air pollutants. Thank you so much.

Comment 8: Table 1 It is not clear whether data in cells belong to 2014 or to 2017. Also the use of the dot “.” is not clear, as the number of digits on the right side of the dot is sometimes 2, 3 or 4 digits.

Response 8: Thank you for this valuable suggestion. We have modified Table 1 in the paper, and united the use of the dot “.”. Please see revised manuscript.

Thank you again for your helping. We really appreciate your feedback.

Round 2

Reviewer 1 Report

Dear Authors,

Thank you very much for your revised manuscript.

The comments are as follows:

1.       Please show the source of the Fig.1. Only if this figure is made yourself, there is no need to show the source.

2.       Please note the significant level of the star in the Table 4.

Author Response

Comment 1: Please show the source of the Fig.1. Only if this figure is made yourself, there is no need to show the source.

Response 1: Thank you for point out the mistake. We have showed the source of the Fig.1 in the paper. Please see the revised manuscript. Thank you so much.

Comment 2: Please note the significant level of the star in the Table 4.

Response 2: Thank you for point out the mistake. We have noted the significant level of the * in the paper. Please see the revised manuscript. Thank you so much.

Thank you again for your helping. We really appreciate your feedback.

Reviewer 2 Report

The introduced changes contributed a lot to increase the overall quality of the paper...

Still, a final gramar review, by a native speaker is highly recommended.

Author Response

Comment 1: Still, a final grammar review, by a native speaker is highly recommended.

Response 1: Thank you for this helpful suggestion. We have asked a native speaker to help us modified the grammar. Please see the revision. Thank you so much.

Thank you again for your helping. We really appreciate your feedback.

Reviewer 3 Report

Please, give a better interpretation of correlations in Table 4, because some of them are not significant, and they lack reliability. Consequently, the explanation associated to them may need to be modified

I recommend you to write a short explanation about the similitudes, differences or interactions among the pollutants considered

Author Response

Comment 1: Please, give a better interpretation of correlations in Table 4, because some of them are not significant, and they lack reliability. Consequently, the explanation associated to them may need to be modified.

Response 1: Thank you for this helpful suggestion. We have further analyzed the results in Section 4.3 of the Discussion, and try to better explain what is shown in Table 4. Please see the Section 4.3. Thank you for your wise advice.

Comment 2: I recommend you to write a short explanation about the similitudes, differences or interactions among the pollutants considered.

Response 2: Thank you for this helpful suggestion. We have written the relationship among the air pollutants in Section 4.2. Please see line 244-246. Thank you for your wise advice.

Thank you again for your helping. We really appreciate your feedback.

Round 3

Reviewer 3 Report

no comment